# Black-box optimization of noisy functions with unknown smoothness

**Jean-Bastien Grill**  **Michal Valko**  **Rémi Munos**
SequeL team, INRIA Lille - Nord Europe, France  Google DeepMind, UK[*]
jean-bastien.grill@inria.fr  michal.valko@inria.fr  munos@google.com

## Abstract

We study the problem of black-box optimization of a function $f$ of any dimension, given function evaluations perturbed by noise. The function is assumed to be locally smooth around one of its global optima, but this *smoothness is unknown*. Our contribution is an adaptive optimization algorithm, POO or *parallel optimistic optimization*, that is able to deal with this setting. POO performs almost as well as the best known algorithms requiring the knowledge of the smoothness. Furthermore, POO works for a larger class of functions than what was previously considered, especially for functions that are *difficult to optimize*, in a very precise sense. We provide a *finite-time* analysis of POO's performance, which shows that its error after $n$ evaluations is at most a factor of $\sqrt{\ln n}$ away from the error of the best known optimization algorithms using the knowledge of the smoothness.

## 1 Introduction

We treat the problem of optimizing a function $f : \mathcal{X} \to \mathbb{R}$ given a finite budget of $n$ noisy evaluations. We consider that the cost of any of these *function evaluations* is high. That means, we care about assessing the optimization performance in terms of the sample complexity, i.e., the number of $n$ function evaluations. This is typically the case when one needs to tune parameters for a complex system seen as a black-box, which performance can only be evaluated by a costly simulation. One such example, is the *hyper-parameter tuning* where the sensitivity to perturbations is large and the derivatives of the objective function with respect to these parameters do not exist or are unknown.

Such setting fits the sequential decision-making setting under *bandit feedback*. In this setting, the actions are the points that lie in a domain $\mathcal{X}$. At each step $t$, an algorithm selects an action $x_t \in \mathcal{X}$ and receives a reward $r_t$, which is a noisy function evaluation such that $r_t = f(x_t) + \varepsilon_t$, where $\varepsilon_t$ is a bounded noise with $\mathbb{E}\left[\varepsilon_t \,|x_t\right] = 0$. After $n$ evaluations, the algorithm outputs its best guess $x(n)$, which can be different from $x_n$. The performance measure we want to minimize is the value of the function at the returned point compared to the optimum, also referred to as *simple regret*,

$$R_n \stackrel{\text{def}}{=} \sup_{x \in \mathcal{X}} f(x) - f\left(x\left(n\right)\right).$$

We assume there exists at least one point $x^\star \in \mathcal{X}$ such that $f(x^\star) = \sup_{x \in \mathcal{X}} f(x)$.

The relationship with bandit settings motivated UCT [10, 8], an empirically successful heuristic that hierarchically partitions domain $\mathcal{X}$ and selects the next point $x_t \in \mathcal{X}$ using upper confidence bounds [1]. The empirical success of UCT on one side but the absence of performance guarantees for it on the other, incited research on similar but theoretically founded algorithms [4, 9, 12, 2, 6].

As the global optimization of the unknown function without absolutely any assumptions would be a daunting needle-in-a-haystack problem, most of the algorithms assume at least a very weak

---

[*]on the leave from SequeL team, INRIA Lille - Nord Europe, France

assumption that the function does *not decrease faster than a known rate* around *one* of its global optima. In other words, they assume a certain *local smoothness* property of $f$. This smoothness is often expressed in the form of a semi-metric $\ell$ that quantifies this regularity [4]. Naturally, this regularity also influences the guarantees that these algorithms are able to furnish. Many of them define a *near-optimality dimension $d$* or a *zooming dimension*. These are $\ell$-dependent quantities used to bound the simple regret $R_n$ or a related notion called *cumulative regret*.

Our work focuses on a notion of such near-optimality dimension $d$ that does not directly relate the smoothness property of $f$ to a specific metric $\ell$ but *directly* to the *hierarchical partitioning* $\mathcal{P} = \{\mathcal{P}_{h,i}\}$, a *tree-based representation* of the space used by the algorithm. Indeed, an interesting fundamental question is to determine a good characterization of the difficulty of the optimization for an algorithm that uses a given hierarchical partitioning of the space $\mathcal{X}$ as its input. The kind of hierarchical partitioning $\{\mathcal{P}_{h,i}\}$ we consider is similar to the ones introduced in prior work: for any depth $h \geq 0$ in the tree representation, the set of *cells* $\{\mathcal{P}_{h,i}\}_{1 \leq i \leq I_h}$ form a partition of $\mathcal{X}$, where $I_h$ is the number of cells at depth $h$. At depth 0, the root of the tree, there is a single cell $\mathcal{P}_{0,1} = \mathcal{X}$. A cell $\mathcal{P}_{h,i}$ of depth $h$ is split into several children subcells $\{\mathcal{P}_{h+1,j}\}_j$ of depth $h+1$. We refer to the standard partitioning as to one where each cell is split into regular same-sized subcells [13].

An important insight, detailed in Section 2, is that a near-optimality dimension $d$ that is independent from the partitioning used by an algorithm (as defined in prior work [4, 9, 2]) *does not embody the optimization difficulty perfectly*. This is easy to see, as for any $f$ we could define a partitioning, perfectly suited for $f$. An example is a partitioning, that at the root splits $\mathcal{X}$ into $\{x^\star\}$ and $\mathcal{X} \setminus x^\star$, which makes the optimization trivial, whatever $d$ is. This insight was already observed by Slivkins [14] and Bull [6], whose *zooming dimension* depends both on the function and the partitioning.

In this paper, we define a notion of near-optimality dimension $d$ which measures the complexity of the optimization problem *directly in terms of the partitioning* used by an algorithm. First, we make the following local smoothness assumption about the function, expressed in terms of the partitioning and *not any metric*: For a given partitioning $\mathcal{P}$, we assume that there exist $\nu > 0$ and $\rho \in (0,1)$, s.t.,

$$\forall h \geq 0, \forall x \in \mathcal{P}_{h,i_h^\star}, \quad f(x) \geq f(x^\star) - \nu\rho^h$$

where $(h, i_h^\star)$ is the (unique) cell of depth $h$ containing $x^\star$. Then, we define the near-optimality dimension $d(\nu, \rho)$ as

$$d(\nu, \rho) \stackrel{\text{def}}{=} \inf\left\{ d' \in \mathbb{R}^+ : \exists C > 0, \forall h \geq 0, \mathcal{N}_h(2\nu\rho^h) \leq C\rho^{-d'h} \right\},$$

where for all $\varepsilon > 0$, $\mathcal{N}_h(\varepsilon)$ is the number of cells $\mathcal{P}_{h,i}$ of depth $h$ s.t. $\sup_{x \in \mathcal{P}_{h,i}} f(x) \geq f(x^\star) - \varepsilon$. Intuitively, functions with smaller $d$ are easier to optimize and we denote $(\nu, \rho)$, for which $d(\nu, \rho)$ is the smallest, as $(\nu_\star, \rho_\star)$. Obviously, $d(\nu, \rho)$ depends on $\mathcal{P}$ and $f$, but *does not depend* on any choice of a specific metric. In Section 2, we argue that this definition of $d$[1] encompasses the optimization complexity *better*. We stress this is not an artifact of our analysis and previous algorithms, such as HOO [4], TaxonomyZoom [14], or HCT [2], can be shown to scale with this new notion of $d$.

Most of the prior bandit-based algorithms proposed for function optimization, for either deterministic or stochastic setting, assume that the smoothness of the optimized function is *known*. This is the case of known *semi-metric* [4, 2] and *pseudo-metric* [9]. This assumption limits the application of these algorithms and opened a very compelling question of whether this knowledge is necessary.

Prior work responded with algorithms not requiring this knowledge. Bubeck et al. [5] provided an algorithm for optimization of Lipschitz functions without the knowledge of the Lipschitz constant. However, they have to assume that $f$ is twice differentiable and a bound on the second order derivative is known. Combes and Proutière [7] treat unimodal $f$ restricted to dimension one. Slivkins [14] considered a general optimization problem embedded in a *taxonomy*[2] and provided guarantees as a function of the *quality* of the taxonomy. The quality refers to the probability of reaching two cells belonging to the same branch that can have values that differ by more that half of the diameter (expressed by the true metric) of the branch. The problem is that the algorithm needs a lower bound on this quality (which can be tiny) and the performance depends inversely on this quantity. Also it assumes that the quality is strictly positive. In this paper, we do not rely on the knowledge of quality and also consider a more general class of functions for which the quality can be 0 (Appendix E).

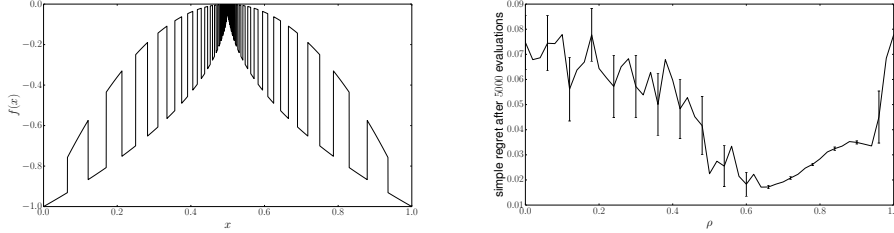

Figure 1: Difficult function $f : x \to s\left(\log_2 |x - 0.5|\right) \cdot \left(\sqrt{|x - 0.5|} - (x - 0.5)^2\right) - \sqrt{|x - 0.5|}$ where, $s(x) = 1$ if the fractional part of $x$, that is, $x - \lfloor x \rfloor$, is in $[0, 0.5]$ and $s(x) = 0$, if it is in $(0.5, 1)$. *Left:* Oscillation between two envelopes of different smoothness leading to a nonzero $d$ for a standard partitioning. *Right:* Regret of HOO after 5000 evaluations for different values of $\rho$.

Another direction has been followed by Munos [11], where in the deterministic case (the function evaluations are not perturbed by noise), their SOO algorithm performs almost as well as the best known algorithms without the knowledge of the function smoothness. SOO was later extended to StoSOO [15] for the stochastic case. However StoSOO only extends SOO for a limited case of *easy instances* of functions for which there exists a semi-metric under which $d = 0$. Also, Bull [6] provided a similar regret bound for the ATB algorithm for a class of functions, called *zooming continuous functions*, which is related to the class of functions for which there exists a semi-metric under which the near-optimality dimension is $d = 0$. But none of the prior work considers a more general class of functions where there is no semi-metric adapted to the standard partitioning for which $d = 0$.

To give an example of a difficult function, consider the function in Figure 1. It possesses a lower and upper envelope around its global optimum that are equivalent to $x^2$ and $\sqrt{x}$; and therefore have different smoothness. Thus, for a standard partitioning, there is no semi-metric of the form $\ell(x, y) = ||x - y||^\alpha$ for which the near-optimality dimension is $d = 0$, as shown by Valko et al. [15]. Other examples of nonzero near-optimality dimension are the functions that for a standard partitioning behave differently depending on the direction, for instance $f : (x, y) \mapsto 1 - |x| - y^2$.

Using a bad value for the $\rho$ parameter can have dramatic consequences on the simple regret. In Figure 1, we show the simple regret after 5000 function evaluations for different values of $\rho$. For the values of $\rho$ that are too low, the algorithm does not explore enough and is stuck in a local maximum while for values of $\rho$ too high the algorithm wastes evaluations by exploring too much.

In this paper, we provide a new algorithm, POO, *parallel optimistic optimization*, which competes with the best algorithms that assume the knowledge of the function smoothness, for a larger class of functions than was previously done. Indeed, POO handles a panoply of functions, including *hard instances*, i.e., such that $d > 0$, like the function illustrated above. We also recover the result of StoSOO and ATB for functions with $d = 0$. In particular, we bound the POO's simple regret as

$$\mathbb{E}[R_n] \leq \mathcal{O}\left(\left(\left(\ln^2 n\right)/n\right)^{1/(2 + d(\nu_\star, \rho_\star))}\right).$$

This result should be compared to the simple regret of the best known algorithm that uses the knowledge of the metric under which the function is smooth, or equivalently $(\nu, \rho)$, which is of the order of $\mathcal{O}((\ln n/n)^{1/(2+d)})$. Thus POO's performance is at most a factor of $(\ln n)^{1/(2+d)}$ away from that of the best known optimization algorithms that require the knowledge of the function smoothness. Interestingly, this factor decreases with the complexity measure $d$: the harder the function to optimize, the less important it is to know its precise smoothness.

## 2 Background and assumptions

### 2.1 Hierarchical optimistic optimization

POO optimizes functions *without* the knowledge of their smoothness using a subroutine, an anytime algorithm optimizing functions *using* the knowledge of their smoothness. In this paper, we use a modified version of HOO [4] as such subroutine. Therefore, we embark with a quick review of HOO.

HOO follows an optimistic strategy close to UCT [10], but unlike UCT, it uses proper confidence bounds to provide theoretical guarantees. HOO refines a partition of the space based on a hierarchical partitioning, where at each step, a yet unexplored cell (a leaf of the corresponding tree) is selected,

and the function is evaluated at a point within this cell. The selected path (from the root to the leaf) is the one that maximizes the minimum value $U_{h,i}(t)$ among all cells of each depth, where the value $U_{h,i}(t)$ of any cell $\mathcal{P}_{h,i}$ is defined as

$$U_{h,i}(t) = \widehat{\mu}_{h,i}(t) + \sqrt{\frac{2\ln(t)}{N_{h,i}(t)}} + \nu\rho^h,$$

where $t$ is the number of evaluations done so far, $\widehat{\mu}_{h,i}(t)$ is the empirical average of all evaluations done within $\mathcal{P}_{h,i}$, and $N_{h,i}(t)$ is the number of them. The second term in the definition of $U_{h,i}(t)$ is a Chernoff-Hoeffding type confidence interval, measuring the estimation error induced by the noise. The third term, $\nu\rho^h$ with $\rho \in (0,1)$ is, by assumption, a bound on the difference $f(x^\star) - f(x)$ for any $x \in \mathcal{P}_{h,i_h^\star}$, a cell containing $x^\star$. Is it this bound, where HOO relies on the knowledge of the smoothness, because the algorithm requires the values of $\nu$ and $\rho$. In the next sections, we clarify the assumptions made by HOO vs. related algorithms and point out the differences with POO.

## 2.2 Assumptions made in prior work

Most of previous work relies on the knowledge of a semi-metric on $\mathcal{X}$ such that the function is either locally smooth near to one of its maxima with respect to this metric [11, 15, 2] or require a stronger, weakly-Lipschitz assumption [4, 12, 2]. Furthermore, Kleinberg et al. [9] assume the full metric. Note, that the semi-metric does not require the triangular inequality to hold. For instance, consider the semi-metric $\ell(x,y) = ||x - y||^\alpha$ on $\mathbb{R}^p$ with $||\cdot||$ being the euclidean metric. When $\alpha < 1$ then this semi-metric does not satisfy the triangular inequality. However, it is a metric for $\alpha \geq 1$. Therefore, using only semi-metric allows us to consider a larger class of functions.

Prior work typically requires two assumptions. The first one is on semi-metric $\ell$ and the function. An example is the *weakly-Lipschitz* assumption needed by Bubeck et al. [4] which requires that

$$\forall x, y \in \mathcal{X}, \quad f(x^\star) - f(y) \leq f(x^\star) - f(x) + \max\left\{f(x^\star) - f(x), \ell(x,y)\right\}.$$

It is a weak version of a Lipschitz condition, restricting $f$ in particular for the values close to $f(x^\star)$.

More recent results [11, 15, 2] assume only a *local smoothness* around one of the function maxima,

$$x \in \mathcal{X} \quad f(x^\star) - f(x) \leq \ell(x^\star, x).$$

The second common assumption *links* the hierarchical partitioning with the semi-metric. It requires the partitioning to be *adapted* to the (semi) metric. More precisely the well-shaped assumption states that there exist $\rho < 1$ and $\nu_1 \geq \nu_2 > 0$, such that for any depth $h \geq 0$ and index $i = 1, \ldots, I_h$, the subset $\mathcal{P}_{h,i}$ is *contained by* and *contains* two open balls of radius $\nu_1\rho^h$ and $\nu_2\rho^h$ respectively, where the balls are w.r.t. the same semi-metric used in the definition of the function smoothness.

'Local smoothness' is weaker than 'weakly Lipschitz' and therefore preferable. Algorithms requiring the local-smoothness assumption always sample a cell $\mathcal{P}_{h,i}$ in a special *representative point* and, in the stochastic case, collect several function evaluations from the same point before splitting the cell. This is not the case of HOO, which allows to sample *any* point inside the selected cell and to expand each cell after one sample. This additional flexibility comes at the price of requiring the stronger weakly-Lipschitzness assumption. Nevertheless, although HOO does not wait before expanding a cell, it does something similar by selecting a path from the root to this leaf that maximizes the minimum of the $U$-value over the cells of the path, as mentioned in Section 2.1. The fact that HOO follows an optimistic strategy even after reaching the cell that possesses the minimal $U$-value along the path is not used in the analysis of the HOO algorithm.

Furthermore, a reason for better dependency on the smoothness in other algorithms, e.g., HCT [2], is not only algorithmic: HCT needs to assume a slightly stronger condition on the cell, i.e., that the single center of the two balls (one that covers and the other one that contains the cell) is actually the same point that HCT uses for sampling. This is stronger than just assuming that there simply exist such centers of the two balls, which are not necessarily the same points where we sample (which is the HOO assumption). Therefore, this is *in contrast with HOO* that samples *any point* from the cell. In fact, it is straightforward to modify HOO to only sample at a representative point in each cell and only require the local-smoothness assumption. In our analysis and the algorithm, we use this modified version of HOO, thereby profiting from this weaker assumption.

Prior work [9, 4, 11, 2, 12] often defined some 'dimension' $d$ of the near-optimal space of $f$ measured according to the (semi-) metric $\ell$. For example, the so-called *near-optimality dimension* [4] measures the size of the near-optimal space $\mathcal{X}_\varepsilon = \{x \in \mathcal{X} : f(x) > f(x^\star) - \varepsilon\}$ in terms of *packing numbers*: For any $c > 0, \varepsilon_0 > 0$, the $(c, \varepsilon_0)$-near-optimality dimension $d$ of $f$ with respect to $\ell$ is defined as

$$\inf \left\{ d \in [0, \infty) : \exists C \text{ s.t. } \forall \varepsilon \leq \varepsilon_0, \mathcal{N}(\mathcal{X}_{c\varepsilon}, \ell, \varepsilon) \leq C\varepsilon^{-d} \right\}, \tag{1}$$

where for any subset $A \subseteq \mathcal{X}$, the packing number $\mathcal{N}(A, \ell, \varepsilon)$ is the maximum number of disjoint balls of radius $\varepsilon$ contained in $A$.

## 2.3 Our assumption

Contrary to the previous approaches, we need *only a single assumption*. We do not introduce any (semi)-metric and instead directly relate $f$ to the hierarchical partitioning $\mathcal{P}$, defined in Section 1. Let $K$ be the maximum number of children cells $(\mathcal{P}_{h+1,j_k})_{1 \leq k \leq K}$ per cell $\mathcal{P}_{h,i}$. We remind the reader that given a global maximum $x^\star$ of $f$, $i_h^\star$ denotes the index of the unique cell of depth $h$ containing $x^\star$, i.e., such that $x^\star \in \mathcal{P}_{h,i_h^\star}$. With this notation we can state our sole assumption on both the partitioning $(\mathcal{P}_{h,i})$ and the function $f$.

**Assumption 1.** *There exists $\nu > 0$ and $\rho \in (0, 1)$ such that*

$$\forall h \geq 0, \forall x \in \mathcal{P}_{h,i_h^\star}, \quad f(x) \geq f(x^\star) - \nu \rho^h.$$

The values $(\nu, \rho)$ defines a lower bound on the possible drop of $f$ near the optimum $x^\star$ according to the partitioning. The choice of the exponential rate $\nu\rho^h$ is made to cover a very large class of functions, as well as to relate to results from prior work. In particular, for a standard partitioning on $\mathbb{R}^p$ and any $\alpha, \beta > 0$, any function $f$ such that $f(x) \sim_{x \to x^\star} \beta ||x - x^\star||^\alpha$ fits this assumption. This is also the case for more complicated functions such as the one illustrated in Figure 1. An example of a function and a partitioning that does not satisfy this assumption is the function $f : x \mapsto 1/\ln x$ and a standard partitioning of $[0, 1)$ because the function decreases too fast around $x^\star = 0$. As observed by Valko [15], this assumption can be weaken to hold only for values of $f$ that are $\eta$-close to $f(x^\star)$ up to an $\eta$-dependent constant in the regret.

Let us note that the set of assumptions made by prior work (Section 2.2) can be reformulated using solely Assumption 1. For example, for any $f(x) \sim_{x \to x^\star} \beta ||x - x^\star||^\alpha$, one could consider the semi-metric $\ell(x, y) = \beta ||x - y||^\alpha$ for which the corresponding near-optimality dimension defined by Equation 1 for a standard partitioning is $d = 0$. Yet we argue that our setting provides a more natural way to describe the complexity of the optimization problem for a given hierarchical partitioning.

Indeed, existing algorithms, that use a hierarchical partitioning of $\mathcal{X}$, like HOO, do not use the full metric information but instead only use the values $\nu$ and $\rho$, paired up with the partitioning. Hence, the precise value of the metric does not impact the algorithms' decisions, neither their performance. What really matters, is how the hierarchical partitioning of $\mathcal{X}$ fits $f$. Indeed, this fit is what we measure. To reinforce this argument, notice again that any function can be trivially optimized given a perfectly adapted partitioning, for instance the one that associates $x^\star$ to one child of the root.

Also, the previous analyses tried to provide performance guaranties based only on the metric and $f$. However, since the metric is assumed to be such that *the cells of the partitioning are well shaped*, the large diversity of possible metrics vanishes. Choosing such metric then comes down to choosing only $\nu$, $\rho$, and a hierarchical decomposition of $\mathcal{X}$. Another way of seeing this is to remark that previous works make an assumption on both the function and the metric, and an other on both the metric and the partitioning. We underline that the metric is actually there just to create a link between the function and the partitioning. By discarding the metric, we merge the two assumptions into a single one and convert a topological problem into a combinatorial one, leading to easier analysis.

To proceed, we define a *new near-optimality dimension*. For any $\nu > 0$ and $\rho \in (0, 1)$, the near-optimality dimension $d(\nu, \rho)$ of $f$ with respect to the partitioning $\mathcal{P}$ is defined as follows.

**Definition 1.** *Near-optimality dimension of $f$ is*

$$d(\rho) \stackrel{\text{def}}{=} \inf \left\{ d' \in \mathbb{R}^+ : \exists C > 0, \ \forall h \geq 0, \ \mathcal{N}_h(2\nu\rho^h) \leq C\rho^{-d'h} \right\}$$

*where $\mathcal{N}_h(\varepsilon)$ is the number of cells $\mathcal{P}_{h,i}$ of depth $h$ such that $\sup_{x \in \mathcal{P}_{h,i}} f(x) \geq f(x^\star) - \varepsilon$.*

The hierarchical decomposition of the space $\mathcal{X}$ is the only prior information available to the algorithm. The (new) near-optimality dimension is a measure of how well is this partitioning adapted to $f$. More precisely, it is a measure of the size of the near-optimal set, i.e., the cells which are such that $\sup_{x \in \mathcal{P}_{h,i}} f(x) \geq f(x^\star) - \varepsilon$. Intuitively, this corresponds to the set of cells that any algorithm would have to sample in order to discover the optimum.

As an example, any $f$ such that $f(x) \sim_{x \to x^\star} ||x - x^\star||^\alpha$, for any $\alpha > 0$, has a zero near-optimality dimension with respect to the standard partitioning and an appropriate choice of $\rho$. As discussed by Valko et al. [15], any function such that the upper and lower envelopes of $f$ near its maximum are of the same order has a near-optimality dimension of zero for a standard partitioning of $[0, 1]$. An example of a function with $d > 0$ for the standard partitioning is in Figure 1. Functions that behave differently in different dimensions have also $d > 0$ for the standard partitioning. Nonetheless, for a some handcrafted partitioning, it is possible to have $d = 0$ even for those troublesome functions.

Under our new assumption and our new definition of near-optimality dimension, one can prove the same regret bound for HOO as Bubeck et al. [4] and the same can be done for other related algorithms.

## 3 The POO algorithm

### 3.1 Description of POO

The POO algorithm uses, as a subroutine, an optimizing algorithm that *requires the knowledge* of the function smoothness. We use HOO [4] as the base algorithm, but other algorithms, such as HCT [2], could be used as well. POO, with pseudocode in Algorithm 1, runs several HOO instances in parallel, hence the name *parallel optimistic optimization*. The number of base HOO instances and other parameters are adapted to the budget of evaluations and are automatically decided on the fly.

Each instance of HOO requires two real numbers $\nu$ and $\rho$. Running HOO parametrized with $(\rho, \nu)$ that are far from the optimal one $(\nu_\star, \rho_\star)$[3] would cause HOO to underperform. Surprisingly, our analysis of this *suboptimality gap* reveals that it does not decrease too fast as we stray away from $(\nu_\star, \rho_\star)$. This motivates the following observation. If we *simultaneously* run a slew of HOOs with different $(\nu, \rho)$s, one of them is going to perform decently well.

In fact, we show that to achieve good performance, we only require $(\ln n)$ HOO instances, where $n$ is the current number of function evaluations. Notice, that we do not require to know the total number of rounds in advance which hints that we can hope for a *naturally anytime* algorithm.

The strategy of POO is quite simple: It consists of running $N$ instances of HOO in parallel, that are all launched with different $(\nu, \rho)$s. At the end of the whole process, POO selects the instance $s^\star$ which performed the best and returns one of the points selected by this instance, chosen uniformly at random. Note that just using a doubling trick in HOO with increasing values of $\rho$ and $\nu$ is not enough to guarantee a good performance. Indeed, it is important to keep track of all HOO instances. Otherwise, the regret rate would suffer way too much from using the value of $\rho$ that is too far from the optimal one.

---

**Algorithm 1** POO

**Parameters:** $K, \mathcal{P} = \{\mathcal{P}_{h,i}\}$
    Optional parameters: $\rho_{\max}, \nu_{\max}$
**Initialization:**
    $D_{\max} \leftarrow \ln K / \ln (1/\rho_{\max})$
    $n \leftarrow 0$ {number of evaluation performed}
    $N \leftarrow 1$ {number of HOO instances}
    $\mathcal{S} \leftarrow \{(\nu_{\max}, \rho_{\max})\}$ {set of HOO instances}
**while** computational budget is available **do**
    **while** $N \geq \frac{1}{2} D_{\max} \ln (n/(\ln n))$ **do**
        **for** $i \leftarrow 1, \ldots, N$ **do** {start new HOOs}
            $s \leftarrow \left(\nu_{\max}, \rho_{\max}^{2N/(2i+1)}\right)$
            $\mathcal{S} \leftarrow \mathcal{S} \cup \{s\}$
            Perform $\frac{n}{N}$ function evaluation with HOO($s$)
            Update the average reward $\widehat{\mu}[s]$ of HOO($s$)
        **end for**
        $n \leftarrow 2n$
        $N \leftarrow 2N$
    **end while**{ensure there is enough HOOs}
    **for** $s \in \mathcal{S}$ **do**
        Perform a function evaluation with HOO($s$)
        Update the average reward $\widehat{\mu}[s]$ of HOO($s$)
    **end for**
    $n \leftarrow n + N$
**end while**
$s^\star \leftarrow \text{argmax}_{s \in \mathcal{S}} \widehat{\mu}[s]$
**Output:** A random point evaluated by HOO($s^\star$)

For clarity, the pseudo-code of Algorithm 1 takes $\rho_{\max}$ and $\nu_{\max}$ as parameters but in Appendix C we show how to set $\rho_{\max}$ and $\nu_{\max}$ *automatically* as functions of the number of evaluations, i.e., $\rho_{\max}(n), \nu_{\max}(n)$. Furthermore, in Appendix D, we explain how to share information between the HOO instances which makes the empirical performance *light-years better*.

Since POO is anytime, the number of instances $N(n)$ is time-dependent and does not need to be known in advance. In fact, $N(n)$ is increased alongside the execution of the algorithm. More precisely, we want to ensure that

$$N(n) \geq \tfrac{1}{2} D_{\max} \ln(n/\ln n), \quad \text{where} \quad D_{\max} \stackrel{\text{def}}{=} (\ln K)/\ln(1/\rho_{\max}).$$

To keep the set of different $(\nu, \rho)$s well distributed, the number of HOOs is not increased one by one but instead is doubled when needed. Moreover, we also require that HOOs run in parallel, perform the same number of function evaluations. Consequently, when we start running new instances, we first ensure to make these instances on par with already existing ones in terms of number of evaluations.

Finally, as our analysis reveals, a good choice of parameters $(\rho_i)$ is not a uniform grid on $[0, 1]$. Instead, as suggested by our analysis, we require that $1/\ln(1/\rho_i)$ is a uniform grid on $[0, 1/(\ln 1/\rho_{\max})]$. As a consequence, we add HOO instances in batches such that $\rho_i = \rho_{\max}^{N/i}$.

## 3.2 Upper bound on POO's regret

POO does not require the knowledge of a $(\nu, \rho)$ verifying Assumption 1 and[4] yet we prove that it achieves a performance close[5] to the one obtained by HOO using the best parameters $(\nu_\star, \rho_\star)$. This result solves the open question of Valko et al. [15], whether the stochastic optimization of $f$ with unknown parameters $(\nu, \rho)$ when $d > 0$ for the standard partitioning is possible.

**Theorem 1.** *Let $R_n$ be the simple regret of POO at step $n$. For any $(\nu, \rho)$ verifying Assumption 1 such that $\nu \leq \nu_{\max}$ and $\rho \leq \rho_{\max}$ there exists $\kappa$ such that for all $n$*

$$\mathbb{E}[R_n] \leq \kappa \cdot \left( \left( \ln^2 n \right) / n \right)^{1/(d(\nu,\rho)+2)}$$

*Moreover, $\kappa = \alpha \cdot D_{\max}(\nu_{\max}/\nu_\star)^{D_{\max}}$, where $\alpha$ is a constant independent of $\rho_{max}$ and $\nu_{max}$.*

We prove Theorem 1 in the Appendix A and B. Notice that Theorem 1 holds for any $\nu \leq \nu_{\max}$ and $\rho \leq \rho_{\max}$ and in particular for the parameters $(\nu_\star, \rho_\star)$ for which $d(\nu, \rho)$ is minimal as long as $\nu_\star \leq \nu_{\max}$ and $\rho_\star \leq \rho_{\max}$. In Appendix C, we show how to make $\rho_{\max}$ and $\nu_{\max}$ *optional*.

To give some intuition on $D_{\max}$, it is easy to prove that it is the attainable upper bound on the near-optimality dimension of functions verifying Assumption 1 with $\rho \leq \rho_{\max}$. Moreover, any function of $[0, 1]^p$, Lipschitz for the Euclidean metric, has $(\ln K)/\ln(1/\rho) = p$ for a standard partitioning.

The POO's performance should be compared to the simple regret of HOO run with the best parameters $\nu_\star$ and $\rho_\star$, which is of order

$$\mathcal{O}\left( \left( (\ln n)/n \right)^{1/(d(\nu_\star,\rho_\star)+2)} \right).$$

Thus POO's performance is only a factor of $\mathcal{O}((\ln n)^{1/(d(\nu_\star,\rho_\star)+2)})$ away from the optimally fitted HOO. Furthermore, we our regret bound for POO is slightly better than the known regret bound for StoSOO [15] in the case when $d(\nu, \rho) = 0$ for the same partitioning, i.e., $\mathbb{E}[R_n] = \mathcal{O}(\ln n/\sqrt{n})$. With our algorithm and analysis, we generalize this bound for any value of $d \geq 0$.

Note that we only give a simple regret bound for POO whereas HOO ensures a bound on both the cumulative and simple regret.[6] Notice that since POO runs several HOOs with non-optimal values of the $(\nu, \rho)$ parameters, this algorithm explores much more than optimally fitted HOO, which dramatically impacts the cumulative regret. As a consequence, our result applies to the simple regret only.

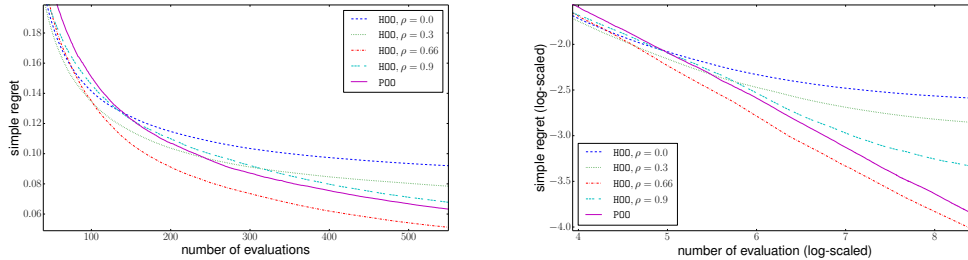

Figure 2: Regret of PO0 and HO0 run for different values of $\rho$.

## 4 Experiments

We ran experiments on the function plotted in Figure 1 for HO0 algorithms with different values of $\rho$ and the PO0[7] algorithm for $\rho_{\max} = 0.9$. This function, as described in Section 1, has an upper and lower envelope that are not of the same order and therefore has $d > 0$ for a standard partitioning.

In Figure 2, we show the simple regret of the algorithms as function of the number of evaluations. In the figure on the left, we plot the simple regret after 500 evaluations. In the right one, we plot the regret after 5000 evaluations in the log-log scale, in order to see the trend better. The HO0 algorithms return a random point chosen uniformly among those evaluated. PO0 does the same for the best empirical instance of HO0. We compare the algorithms according to the expected simple regret, which is the difference between the optimum and the expected value of function value at the point they return. We compute it as the average of the value of the function for all evaluated points. While we did not investigate possibly different heuristics, we believe that returning the deepest evaluated point would give a better empirical performance.

As expected, the HO0 algorithms using values of $\rho$ that are too low, do not explore enough and become quickly stuck in a local optimum. This is the case for both UCT (HO0 run for $\rho = 0$) and HO0 run for $\rho = 0.3$. The HO0 algorithm using $\rho$ that is too high waste their budget on exploring too much. This way, we empirically confirmed that the performance of the HO0 algorithm is greatly impacted by the choice of this $\rho$ parameter for the function we considered. In particular, at $T = 500$, the empirical regret of HO0 with $\rho = 0.66$ was a half of the regret of UCT.

In our experiments, HO0 with $\rho = 0.66$ performed the best which is a bit lower than what the theory would suggest, since $\rho_\star = 1/\sqrt{2} \approx 0.7$. The performance of HO0 using this parameter is almost matched by PO0. This is surprising, considering the fact the PO0 was simultaneously running 100 different HO0s. It shows that carefully sharing information between the instances of HO0, as described and justified in Appendix D, has a major impact on empirical performance. Indeed, among the 100 HO0 instances, only two (on average) actually needed a fresh function evaluation, the 98 could reuse the ones performed by another HO0 instance.

## 5 Conclusion

We introduced PO0 for global optimization of stochastic functions with unknown smoothness and showed that it competes with the best known optimization algorithms that know this smoothness. This results extends the previous work of Valko et al. [15], which is only able to deal with a near-optimality dimension $d = 0$. PO0 is provable able to deal with a trove of functions for which $d \geq 0$ for a standard partitioning. Furthermore, we gave a new insight on several assumptions required by prior work and provided a more natural measure of the complexity of optimizing a function given a hierarchical partitioning of the space, without relying on any (semi-)metric.

**Acknowledgements** The research presented in this paper was supported by French Ministry of Higher Education and Research, Nord-Pas-de-Calais Regional Council, a doctoral grant of École Normale Supérieure in Paris, Inria and Carnegie Mellon University associated-team project EduBand, and French National Research Agency project ExTra-Learn (n.ANR-14-CE24-0010-01).

## Footnotes

[1] we use the simplified notation $d$ instead of $d(\nu, \rho)$ for clarity when no confusion is possible

[2] which is similar to the hierarchical partitioning previously defined

[3]the parameters $(\nu, \rho)$ satisfying Assumption 1 for which $d(\nu, \rho)$ is the smallest

[4] note that several possible values of those parameters are possible for the same function

[5] up to a logarithmic term $\sqrt{\ln n}$ in the simple regret

[6] in fact, the bound on the simple regret is a direct consequence of the bound on the cumulative regret [3]

[7]code available at https://sequel.lille.inria.fr/Software/PO0

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
