[Supplementary Material]

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

 `HOO` using this parameter is almost matched by `POO`. This is surprising, considering the fact the `POO` was simultaneously running 100 different `HOO`s. It shows that carefully sharing information between the instances of `HOO`, as described and justified in Appendix D, has a major impact on empirical performance. Indeed, among the 100 `HOO` instances, only two (on average) actually needed a fresh function evaluation, the 98 could reuse the ones performed by another `HOO` instance.

## 5 Conclusion

We introduced `POO` for global optimization of stochastic functions with unknown smoothness and showed that it competes with the best known optimization algorithms that know this smoothness. This results extends the previous work of Valko et al. [15], which is only able to deal with a near-optimality dimension $d = 0$. `POO` is provable able to deal with a trove of functions for which $d \geq 0$ for a standard partitioning. Furthermore, we gave a new insight on several assumptions required by prior work and provided a more natural measure of the complexity of optimizing a function given a hierarchical partitioning of the space, without relying on any (semi-)metric.

**Acknowledgements**  The research presented in this paper was supported by French Ministry of Higher Education and Research, Nord-Pas-de-Calais Regional Council, a doctoral grant of École Normale Supérieure in Paris, Inria and Carnegie Mellon University associated-team project EduBand, and French National Research Agency project ExTra-Learn (n.ANR-14-CE24-0010-01).

## Footnotes

*on the leave from SequeL team, INRIA Lille - Nord Europe, France

[1]we use the simplified notation $d$ instead of $d(\nu, \rho)$ for clarity when no confusion is possible

[2]which is similar to the hierarchical partitioning previously defined

[3] the parameters $(\nu, \rho)$ satisfying Assumption 1 for which $d(\nu, \rho)$ is the smallest

[4]note that several possible values of those parameters are possible for the same function

[5]up to a logarithmic term $\sqrt{\ln n}$ in the simple regret

[6]in fact, the bound on the simple regret is a direct consequence of the bound on the cumulative regret [3]

[7]code available at https://sequel.lille.inria.fr/Software/POO

[8]even though it scales only as $\ln n$ with the number of evaluations $n$, it does not scale well with $\rho_{\max}$

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

## A  Proof sketch of Theorem 1

In this part we give the roadmap of the proof. The full proof is in Appendix B.

**First step**  For any choice of $\rho_\star$ verifying Assumption 1 and any suboptimal $\rho$ such that

$$0 < \rho_\star \leq \rho < 1,$$

we bound the difference of near-optimality dimension

$$d\left(\rho\right) - d\left(\rho_\star\right) \leq \ln K \left( \frac{1}{\ln\left(1/\rho\right)} - \frac{1}{\ln\left(1/\rho_\star\right)} \right).$$

And deduce that

$$\min_{i:\rho_i \geq \rho_\star} [d(\rho_i) - d(\rho_\star)] \leq \frac{D_{\max}}{N},$$

**Second step**  By simultaneously running a large number of HOO instances, we ensure that for all $\rho_\star \leq \rho_{\max}$, one of them uses a $\rho$ close to $\rho_\star$ and therefore suffers a low regret. On the other hand, simultaneously running a large number of HOOs has a cost, as more evaluations need to be done at each step, one for each HOO. We optimize this tradeoff to deduce the following good choice of $\delta$, which is the maximum distance $|d\left(\rho_i\right) - d(\rho_j)|$, where $i$ and $j$ are two consecutive HOOs.

$$\delta = \mathcal{O}\left(\ln\left(t/\ln t\right)\right).$$

**Third step**  Using the result of the second step, we can compute the simple regret $R_n^\rho$ of the HOO instance running with the parameter $\bar{\rho} > \rho_\star$, which is the closest to $\rho_\star$. Note that, as POO is running, the instance it choose may change over time and so $\bar{\rho}$ depends on $n$.

We prove that there exists a constant $\alpha > 0$ such that for all $n$, $\nu_{\max} > 0$, and $\rho_{\max} < 1$,

$$R_n^\rho \leq \alpha \cdot D_{\max}(\nu_{\max}/\nu_\star)^{D_{\max}} \left(\left(\ln^2 n\right)/n\right)^{1/(d(\bar{\rho})+2)}.$$

**Fourth step**  At the end of the algorithm, we empirically determine which HOO performed the best. However, this best empirical instance may not be the instance running with $\rho$ closest to the optimal unknown $\rho_\star$. Nonetheless, we prove that this error is small enough such that it only impacts the simple regret by a constant factor.

## B  Full proof of Theorem 1

### B.1  First step

We show that for any choice of $\rho_\star$ verifying Assumption 1 and any $\rho$ such that $0 < \rho_\star \leq \rho < 1$,

$$d\left(\rho\right) - d\left(\rho_\star\right) \leq \ln K \left( \frac{1}{\ln\left(1/\rho\right)} - \frac{1}{\ln\left(1/\rho_\star\right)} \right).$$

We start by defining $\mathcal{I}_h(\varepsilon)$ as the set of cells of depth $h$ which are $\varepsilon$-near-optimal,

$$\mathcal{I}_h\left(\varepsilon\right) \stackrel{\text{def}}{=} \left\{ i : \sup_{x \in \mathcal{P}_{h,i}} f(x) \geq f(x_\star) - \varepsilon \right\}.$$

$\mathcal{N}_h(\varepsilon)$, defined in Section 1, is then equal to the cardinality of $\mathcal{I}_h(\varepsilon)$. Notice that if a cell $(h, i)$ is $\varepsilon$-near-optimal then all of its antecedents are also $\varepsilon$-near-optimal. Therefore, for any $\varepsilon$ and $h' > h$, the cells in $\mathcal{I}_{h'}(\varepsilon)$ are descendants of the cells in $\mathcal{I}_h(\varepsilon)$.

Since the number of descendants at depth $h'$ of a cell at depth $h' > h$ is bounded by $K^{h'-h}$ we bound the cardinality $\mathcal{N}_h(\varepsilon)$ of $\mathcal{I}_{h'}(\varepsilon)$,

$$\forall\varepsilon, \forall h' > h, \quad \mathcal{N}_{h'}(\varepsilon) \leq K^{h'-h}\mathcal{N}_h(\varepsilon).$$

By definition of the near-optimality dimension, we know that for any $\nu > 0$ and $\rho_\star \in (0,1)$, there exists $C$ such that for all $h$,

$$\mathcal{N}_h\left(2\nu\rho^h\right) \leq C\rho^{-d(\rho)h}.$$

We define $C(\nu, \rho)$ as the smallest $C$ verifying the above condition.

For any $0 < \nu_\star < \nu$, $0 < \rho_\star < \rho < 1$ and any integer $h \geq h_{\min} \stackrel{\text{def}}{=} \ln(\nu/\nu_\star)/\ln(1/\rho)$ let us define $h_\star$ as the greatest integer such that $\nu\rho^h < \nu_\star\rho_\star^{h_\star}$. From this definition, we get $\nu\rho^h \geq \nu_\star\rho_\star^{h_\star+1}$ from which we deduce that

$$h_\star \geq h \cdot \frac{\ln\rho}{\ln\rho_\star} + \frac{\ln\nu - \ln\nu_\star}{\ln\rho_\star} - 1,$$

and then

$$h - h_\star \leq h_\star \ln\rho_\star \left(\frac{1}{\ln\rho} - \frac{1}{\ln\rho_\star}\right) + \frac{\ln\rho_\star + \ln\nu_\star - \ln\nu}{\ln\rho}.$$

Since $\mathcal{N}_h(\varepsilon)$ is not increasing in $\varepsilon$, $\nu\rho^h < \nu_\star\rho_\star^{h_\star}$ implies

$$\mathcal{N}_h(2\nu\rho^h) \leq \mathcal{N}_h(2\nu_\star\rho_\star^{h_\star}).$$

We now put everything together to obtain

$$\begin{aligned}
\mathcal{N}_h(2\nu\rho^h) &\leq \mathcal{N}_h(2\nu_\star\rho_\star^{h_\star}) \\
&\leq K^{h-h_\star}\mathcal{N}_{h_\star}(2\nu_\star\rho_\star^{h_\star}) \\
&\leq K^{(\ln\rho_\star+\ln\nu_\star-\ln\nu)/\ln\rho+h_\star\ln\rho_\star(1/\ln\rho-1/\ln\rho_\star)}C(\nu_\star,\rho_\star)\rho_\star^{-d(\rho_\star)h_\star} \\
&\leq C(\nu_\star,\rho_\star)K^{(\ln\rho_\star+\ln\nu_\star-\ln\nu)/\ln\rho}\rho_\star^{-h_\star[d(\rho_\star)+\ln K(1/\ln(1/\rho)-1/\ln(1/\rho_\star))]}.
\end{aligned}$$

From $\nu\rho^h < \nu_\star\rho_\star^{h_\star}$ and $\nu_\star < \nu$ we get $\rho^{-h} > \rho_\star^{-h_\star}$ and therefore

$$\mathcal{N}_h(2\nu\rho^h) \leq C(\nu_\star,\rho_\star)K^{(\ln\rho_\star+\ln\nu_\star-\ln\nu)/\ln\rho}\rho^{-h[d(\rho_\star)+\ln K(1/\ln(1/\rho)-1/\ln(1/\rho_\star))]}.$$

We just proved that there exists $C$ such that for all $h > 0$

$$\mathcal{N}_h(2\nu\rho^h) \leq C\rho^{-h[d(\rho_\star)+\ln K(1/\ln(1/\rho)-1/\ln(1/\rho_\star))]}.$$

By taking

$$C \stackrel{\text{def}}{=} \max\left(C(\nu_\star,\rho_\star)K^{(\ln\rho_\star+\ln\nu_\star-\ln\nu)/\ln\rho}, K^{h_{\min}}\right),$$

we deduce by the definition of the near-optimality dimension the following bound

$$d(\rho) \leq d(\rho_\star) + \ln K\left(\frac{1}{\ln(1/\rho)} - \frac{1}{\ln(1/\rho_\star)}\right).$$

We can now deduce that POO should use $\rho_i$ parameters that satisfy

$$\frac{1}{\ln(1/\rho_i)} = \frac{i}{N}\frac{1}{\ln(1/\rho_{\max})},$$

where $N$ is the total number of HOO instances run and $i \in \{1, \ldots, N\}$.

We now define $\bar{\rho}$ as the closest $\rho_i$ to $\rho_\star$ used by an existing HOO instance, such that $\rho_i > \rho_\star$.

$$\bar{\rho} \stackrel{\text{def}}{=} \operatorname*{arg\,min}_{i:\rho_i \geq \rho_\star}\left[d(\rho_i) - d(\rho_\star)\right].$$

Since we assumed that $\rho_\star < \rho_{\max}$, we know that

$$d(\bar{\rho}) - d(\rho_\star) \leq \frac{D_{\max}}{N}, \quad \text{with} \quad D_{\max} \stackrel{\text{def}}{=} (\ln K)/\ln(1/\rho_{\max}).$$

## B.2 Second step

Let us now compute the optimal number of $N$ instances to run in parallel. We bound the logarithm of the cumulative regret $R_t^{\nu,\rho}$ of a single HOO instance using parameters $\nu$ and $\rho$ after this particular instance performed $t$ function evaluations. In particular, we bound the regret by a linear approximation for $\rho \sim \rho_\star$. In the following, $\beta$ is a numerical constant coming from the analysis of HOO [4]. For all $t > 0$, we have

$$
\begin{aligned}
\ln R_t^{\nu,\rho} &\leq \ln \beta + \frac{\ln C(\nu,\rho)}{2 + d(\rho)} - \frac{\ln(t/\ln t)}{2 + d(\rho)} \\
&= \ln \beta + \frac{\ln C(\nu,\rho)}{2 + d(\rho)} - \frac{\ln(t/\ln t)}{2 + d(\rho_\star)} \cdot \frac{1}{1 + (d(\rho) - d(\rho_\star))/(2 + d(\rho_\star))} \\
&\leq \ln \beta + \frac{\ln C(\nu,\rho)}{2 + d(\rho)} - \frac{\ln(t/\ln t)}{2 + d(\rho_\star)} \cdot \left(1 - \frac{d(\rho) - d(\rho_\star)}{2 + d(\rho_\star)}\right).
\end{aligned}
$$

After $n$ function evaluations by POO, each instance performed at least $t = \lfloor n/N \rfloor$ function evaluations. We can now bound the commutative regret $R_n^{\text{POO},\nu,\bar\rho}$ of the HOO instance using $\nu$ and $\bar\rho$ after $n$ evaluations performed by all the instances

$$
\ln R_n^{\text{POO},\nu,\bar\rho} \leq \ln \beta + \frac{\ln C(\nu,\bar\rho)}{2 + d(\bar\rho)} + \ln \left(\frac{\ln \lfloor n/N \rfloor}{\lfloor n/N \rfloor}\right) \left(\frac{1}{2 + d(\rho_\star)} - \frac{D_{\max}/N}{(2 + d(\rho_\star))^2}\right). \tag{2}
$$

Optimizing this upper bound for $N$ leads to the following choice of $N$,

$$
N \sim \tfrac{1}{2} D_{\max} \ln(n/\ln n).
$$

Therefore, in POO we choose to ensure $N \geq \frac{1}{2} D_{\max} \ln(n/\ln n)$.

If the time horizon was known in advance, $N$ could be any integer. Nevertheless, since the algorithm is anytime, all the previous HOO instances have to be kept and new instances need to be added in between. Therefore, we restrict $N$ to be of the form $2^i$, for $i \in \mathbb{N}$.

As a consequence of this choice, $N$ can be at most 2 times its lower bound and therefore

$$
\tfrac{1}{2} D_{\max} \ln(n/\ln n) \leq N \leq D_{\max} \ln(n/\ln n).
$$

## B.3 Third step

Using our choice of $N$, we can bound the regret of the HOO instance using $\bar\rho$. We proceed by separately bounding each of the terms in Equation 2.

$$
\begin{aligned}
\frac{\ln C(\nu,\bar\rho)}{2 + d(\bar\rho)} &\leq \frac{1}{2 + d(\rho_\star)} \ln C(\nu,\bar\rho) \\
&\leq \frac{1}{2 + d(\rho_\star)} \ln \max \left(C(\nu_\star,\rho_\star) K^{(\ln \rho_\star + \ln \nu_\star - \ln \nu)/\ln \bar\rho}, K^{h_{\min}}\right) \\
&\leq \frac{1}{2 + d(\rho_\star)} \max \left(\ln C(\nu_\star \rho_\star) + \ln K \left(\frac{\ln 1/\rho_\star}{\ln 1/\bar\rho} + \frac{\ln(\nu/\nu_\star)}{\ln 1/\rho}\right), \ln \left[K^{\ln(\nu/\nu_\star)/\ln(1/\rho)}\right]\right) \\
&\leq \frac{1}{2 + d(\rho_\star)} \max \left(\ln C(\nu_\star \rho_\star) + \max \left(\frac{\ln K \ln \rho_\star D_{\max}}{N}, 2\right) + \frac{\ln K \ln \frac{\nu_{\max}}{\nu_\star}}{\ln 1/\rho}, D_{\max} \ln \frac{\nu}{\nu_\star}\right) \\
&\leq \gamma + \frac{D_{\max}}{2 + d(\rho_\star)} \ln(\nu_{\max}/\nu_\star)
\end{aligned}
$$

In the last expression, $\gamma$ is a quantity independent of $\nu_{\max}$, $\rho_{\max}$, and $N$.

We now use $N \leq D_{\max} \ln(n/\ln n)$ to get

$$
\ln \left(\frac{\ln \lfloor n/N \rfloor}{\lfloor n/N \rfloor}\right) \leq \ln \left(D_{\max} \ln n \ln(n/\ln n)/n\right).
$$

To bound the last term, we use $\frac{1}{2}D_{\max}\ln(n/\ln n) \leq N$ to get

$$-\ln\left(\frac{\ln\lfloor n/N\rfloor}{\lfloor n/N\rfloor}\right)\frac{D_{\max}/N}{(2+d(\rho_\star))^2} \leq \ln\left(\frac{1}{D_{\max}}\cdot\frac{n}{\ln n}\cdot\frac{1}{\ln(n/\ln n)}\right)\frac{1}{2\ln(n/\ln n)} \leq 2.$$

We can finally bound the regret $R_n^{\text{POO},\bar\rho}$ of the HOO instance using $\bar\rho$ after $n$ function evaluations overall. Combining the results above, we know that for all $n$, $\nu_{\max}$, and $\rho_{\max}$,

$$R_n^{\text{POO},\bar\rho} \leq \beta\exp(\gamma+2)\left(D_{\max}\left(\nu_{\max}/\nu_\star\right)^{D_{\max}}(\ln n)\ln(n/\ln n)/n\right)^{1/(2+d(\bar\rho))}.$$

We bound $\ln(n/\ln n)$ by $\ln n$ to get the following bound, there exists $\alpha$ independent of $\rho_{\max}$ and $\nu_{\max}$ such that

$$R_n^{\text{POO},\bar\rho} \leq \alpha\cdot D_{\max}(\nu_{\max}/\nu_\star)^{D_{\max}}\left((\ln^2 n)/n\right)^{1/(d(\bar\rho)+2)}.$$

### B.4 Fourth step

Let $(X_{i,j})_{i\leq n,j\leq N}$ be a family of points in $\mathcal{X}$ evaluated by POO. We denote $\widehat{f}(X_{i,j})$ the noisy evaluation at $X_{i,j}$ and $f(X_{i,j}) = \mathbb{E}[\widehat{f}(X_{i,j})]$. We also define:

$$\mu_j \stackrel{\text{def}}{=} \frac{1}{n}\sum_{i=1}^n f(X_{i,j}) \qquad \widehat{\mu}_j \stackrel{\text{def}}{=} \frac{1}{n}\sum_{i=1}^n \widehat{f}(X_{i,j})$$

$$j \stackrel{\text{def}}{=} \underset{1\leq j\leq N}{\arg\max}\,\mu_j \qquad \widehat{j} \stackrel{\text{def}}{=} \underset{1\leq j\leq N}{\arg\max}\,\widehat{\mu}_j$$

By Hoeffding-Azuma inequality for martingale differences, for any $\Delta > 0$,

$$\mathbb{P}\left[\sum_{i=1}^n \widehat{f}(X_{i,j}) - f(X_{i,j}) > n\Delta\right] \leq \exp\left(-\frac{2(n\Delta)^2}{n}\right).$$

Therefore

$$\mathbb{P}[\widehat{\mu}_j - \mu_j > \Delta] \leq \exp\left(-2n\Delta^2\right).$$

As we have

$$\forall x \geq 0, x\cdot\exp\left(-2nx^2\right) \leq \frac{e^{-2}}{2\sqrt{n}},$$

we can now integrate $\exp\left(-2n\Delta^2\right)$ over $\Delta \in [0,1]$ to get

$$\mathbb{E}[\widehat{\mu}_j - \mu_j] \leq \frac{e^{-2}}{2\sqrt{n}}.$$

Now consider

$$\mathbb{E}\left[\mu_j - \mu_{\widehat{j}}\right] = \mathbb{E}\left[\mu_j - \widehat{\mu}_j\right] + \mathbb{E}\left[\widehat{\mu}_j - \widehat{\mu}_{\widehat{j}}\right] + \mathbb{E}\left[\widehat{\mu}_{\widehat{j}} - \mu_{\widehat{j}}\right].$$

Notice that the first and last term are both bounded by $e^{-2}/(2\sqrt{n})$ and the middle term is negative. Finally, taking a union bound over the $N$ variables $\mu_j$ we get

$$\mathbb{E}\left[\mu_{j_\star} - \mu_{\widehat{j}_\star}\right] \leq \frac{e^{-2}N}{\sqrt{n}}.$$

As $N = o(\ln n)$, we conclude that this additional term is negligible with respect to

$$(\ln n\ln(n/\ln n)/n)^{1/(2+d(\rho_\star))}.$$

## C  Increasing sequence for $\rho_{\max}$ and $\nu_{\max}$

Besides the number $K$ of children for each cell, POO needs two parameters, $\rho_{\max} \in (0, 1)$ and $\nu_{\max} > 0$. Theorem 1 states that POO run with those parameters performs almost as well as the best instance of HOO run with $\nu \le \nu_{\max}$ and $\rho \le \rho_{\max}$, i.e., corresponding to the near-optimality dimension $\min\{d(\nu, \rho), \nu \le \nu_{\max}, \rho \le \rho_{\max}\}$.

Therefore, the larger the values $\rho_{\max}$ and $\nu_{\max}$ used by POO, the wider the set of HOO instances that we can compete with. Nevertheless, large values of $\rho_{\max}$ and $\nu_{\max}$ impact the performance by a multiplicative constant of order $D_{\max}\nu_{\max}{}^{D_{\max}}$. This tradeoff between performance and size of our comparison class is unfortunate but unavoidable.

In practice, as we strive for an algorithm that does not require the knowledge of the smoothness we may increase the values of $\rho_{\max}(n)$ and $\nu_{\max}(n)$ with the number of evaluations $n$, so that the class of functions covered by POO gets bigger with the numerical budget. Nevertheless, the increase should be slow enough so that we do not compromise the performance. In particular, we will require that $\nu_{\max}(n)^{D_{\max}(n)}$ does not increase too fast. In fact, any sequence $\rho_{\max}(n)$ converging to 1 and $\nu_{\max}(n)$ diverging to infinity impacts the regret by an additive term which is the smallest time $n$ such that $\rho^\star < \rho_{\max}(n)$ and $\nu^\star < \nu_{\max}(n)$, i.e., the first time the assumptions are verified. A slowly increasing sequence means a smaller impact on the regret rate but a higher additive term (a constant independent of $n$). Any sensible choice of increasing sequence $\rho_{\max}(n)$ and $\nu_{\max}(n)$, impacting the rate by only a subpolynomial factor, is a valid choice.

Algorithm 1 is described using constant $\rho_{\max}$ and $\nu_{\max}$ for clarity, but its implementation is easily modifiable to deal with increasing values of these two parameters while preserving the anytime property of the algorithm, as follows. At any time, all the HOO instances must use the same $\nu_{\max}$ parameter. On the other hand, considering $\rho_{\max}$, the value of $D_{\max}$ has to be increased such that the already running HOO instances stay relevant. One way to do that is to increase $D_{\max}$ as $D_{\max}(N+1)/N$ and run an additional HOO instance. An alternative solution is to perform, each time when needed, the following increment $\rho_{\max} \leftarrow \sqrt{\rho_{\max}}$ and run $N$ additional HOO instances with parameters $\rho_{\max}{}^{2N/i}$, for $i \in \{1, \ldots, N\}$.

## D  Information sharing among parallel runs

Since we run several instances of HOO on the same partitioning of $\mathcal{X}$, we may think of *sharing the samples* among them, in order to decrease the estimation error. However, this needs to be done carefully in order to avoid adding unwanted bias in the estimation of the $U$ values in the HOO instances. Ideally, each HOO instance would reuse all function evaluations acquired by all other instances. Unfortunately, this solution would not easily come with theoretical guaranties, as this would reduce artificially the confidence intervals at some cells and introduce *search bias*.

Instead, whenever a HOO instance requires a function evaluation, we perform a *look-up* to find out whether another HOO instance has already evaluated $f$ at this point. In affirmative, then instead of evaluating the function at this point again, we simply *reuse* the sample. This way, HOO instances are *not given access to samples they never asked for*. However, the empirical regrets of HOOs becomes correlated with each other. This is not a problem because in B.4, we do not assume the independence between empirical means of HOOs, only the independence of rewards within each instance—which still holds. Therefore with this modification, our theoretical guaranties continue to apply. Note that if all the instances share all their rewards, then they are all equivalent and there is no mistake possible. Then one can show, that the worst case is when no rewards are shared and the error due to choosing the wrong instance actually decreases when the information is shared.

Finally, we want to stress that sharing information is extremely important in practice, as our experiments reveal. Since the number of HOO instances can be very large[8] one could expect the performance of POO to be pitiful. However, as the vast majority of the function evaluations are in practice shared, POO performs almost as well as HOO fitted with the best parameters. Summing up, although the performance bound on the simple regret with this modification is the same, empirical performance *improves tremendously*.

# E  Zero-quality functions

For any $\rho \in (0, 1)$, we construct a locally Lipschitz function with a rate $\rho$ and a constant $\nu = 1$ that POO can provably optimize and its *quality*, as defined by Definition 2, is zero. In order to properly define the quality, we use the uniform distribution on $[0, 1]$ to sample from a node of the partitioning.

**Definition 2** (Slivkins [14]). *The quality is the largest $q \in (0, 1)$ such that for each subtree $v$ containing the optimum, there exist nodes $u$ and $u'$ such that $\mathbb{P}(u|v)$ and $\mathbb{P}(u'|v)$ are at least $q$ and*

$$|f(u) - f(u')| \geq \tfrac{1}{2} \sup_{x, y \in v} |f(x) - f(y)|.$$

We construct such function $f$ on the interval $[0, 1]$, its maximum being attained in $x^\star = 0$ with $f(0) = 0$. For any $x \neq 0$ we define $f$ as follows. For any $h \geq 0$ we define $f$ on $\left(\frac{1}{2^{h+1}}, \frac{1}{2^h}\right]$ as

$$\forall x \in \left(\frac{1}{2^{h+1}}, \frac{1 + 1/(h+1)}{2^{h+1}}\right], \quad f(x) = -\rho^h,$$

$$\forall x \in \left(\frac{1 + 1/(h+1)}{2^{h+1}}, \frac{1}{2^h}\right], \quad f(x) = -\frac{\rho^h}{3}.$$

We also consider the standard partitioning on $[0, 1]$.

The optimal node of depth $h$ corresponds to the interval $\left[0, 2^{-h}\right]$. By our definition of $f$,

$$\forall x \in \left[0, 2^{-h}\right], \quad f(0) - f(x) \leq \rho^h$$

$$f(0) - f\left(\frac{1 + 1/(h+1)}{2^{h+1}}\right) = \rho^h,$$

from which we conclude that $f$ is locally Lipschitz with rate $\rho$ and therefore can be optimized by POO with provable finite-time guarantees (Theorem 1).

Now we prove that the quality of this function is zero. Using Definition 2, we can do it by showing that there exists no such $q \in (0, 1)$, for which there could be a node $v$ along the optimal path with $u$ and $u'$ verifying $\mathbb{P}(u|v) \geq q$ (and same for $u'$) such that

$$\sup_{x \in u} f(x) - \sup_{x \in u'} f(x) \geq \sup_{x \in v} \frac{f(0) - f(x)}{2}. \tag{3}$$

Let $q$ be a real number from $(0, 1)$ and consider any $h > 1/q$. We pick $v = \left[0, 2^{-h}\right]$.

$$\mathbb{P}\left(\left\{x \in v : f(x) \leq -\frac{\rho^h}{2}\right\}\middle|v\right) =$$

$$= \frac{1}{2}\mathbb{P}\left(\left\{x \in \left[0, \frac{1}{2^{h+1}}\right] : f(x) \leq -\frac{\rho^h}{2}\right\}\middle|v\right) + \frac{1}{2}\mathbb{P}\left(\left\{x \in \left[\frac{1}{2^{h+1}}, \frac{1}{2^h}\right] : f(x) \leq -\frac{\rho^h}{2}\right\}\middle|v\right)$$

$$= \frac{1}{2}\mathbb{P}\left(\left\{x \in \left[0, \frac{1}{2^{h+1}}\right] : f(x) \leq -\frac{\rho^h}{2}\right\}\middle|v\right) + \frac{1}{2(h+1)}$$

$$\leq \frac{1}{2}\sum_{k=1}^{\infty} \frac{1}{2^k(h+k+1)} + \frac{1}{2(h+1)}$$

$$\leq \frac{1}{h+1} < q$$

Notice that if $u'$ verifies (3), then $u'$ is included in $\left\{x \in v : f(x) \leq -\rho^h/2\right\}$. Combined with the equation above, we have that

$$\mathbb{P}(u|v) \leq \mathbb{P}\left(\left\{x \in v : f(x) \leq -\rho^h/2\right\}\middle|v\right) < q,$$

which is a contradiction. Since this holds for any $q > 0$, we deduce that the quality of $f$ is zero. Yet $f$ is Lipschitz with rate $\rho \in (0, 1)$ and therefore $f$ can be optimized by POO.