[Reviews · NeurIPS 2015]

Submitted by Assigned_Reviewer_1

The paper is about optimization of an unknown function with bandit feedback. The contribution of the paper is twofold: 1. it introduces a notion named near-optimality dimension with respect to the hierarchical partition that directly measures the hardness of the optimization problem. This contrasts with making two separate assumptions in the existing literature. 2. It proposes POO, an adaptive variant of HOO algorithm, which does not require parameters \nu and \rho (measuring the property of the unknown function f) to be known in advance. POO only pays an additional O(ln n) factor in convergence rate compared to HOO.

Quality: the paper is technically correct as far as I read.

Clarity: the paper is quite well-written, the relevant notion are introduced naturally.

Originality: I would think the first contribution is fairly standard, since this is based on a fusion of conditions for existing algorithm to work. The second contribution is based on an application of ``doubling trick" and guessing the possible parameters in parallel, which does not appear surprising.

Significance: the paper contains some interesting observations, and resolves an open question posed in [13]. But my concern is that the techniques employed appear to be quite standard.

EDIT: After reading the rebuttal and discussion among reviewers, I still think the technique of "guessing" the unknown parameters is fairly standard. For example, in the active learning literature, there has been several works using this trick, making the query complexity dependent on some niceness property of the data distribution. Specifically, Section 3.2 (pages 9-11) and 4.1 of [1] discusses how to make the algorithm adaptive to the unknown error of the best classifier/adaptive to the bounded noise rate. Section 6, page 11 of [2] also deals with this issue, which is not the main contribution of the paper.

[1] M-F. Balcan, S. Hanneke. Robust Interactive Learning. COLT 2012. Available at Arxiv: 1111.1422.

[2] S. Hanneke. Teaching Dimension and the Complexity of Active Learning. COLT 2007.
Summary: The paper is about optimization of an unknown function with bandit feedback. It provides some interesting observations and answers an open question in the literature, but the techniques are not surprising.

Submitted by Assigned_Reviewer_2

A lot of acronyms are used without inroducing them, e.g. UCT, HOO, SOO, StoSOO, ATB, which makes the paper impossible to read without reading all the references. In general, I recommend that the authors revise the paper thoroughly for clarity and understandability.
Summary: This paper proposes a parallel approach to cope with unknown smoothness in noisy black box optimization. The idea is very interesting. To my assessment, such an approach has rather limited practical value, since noisy black box optimzation typically poses the challenge of expensive function evaluations. In these cases, O(...) theorems have little meaning.

Submitted by Assigned_Reviewer_3

The problem of bandit optimization of a noisy function without an explicit knowledge of smoothness is considered.

The paper introduces a new variant of

the existing HOO algorithm to solve this problem. The main idea is that to run N

instances HOO in parallel for different choice of smoothness and then to choose the instance with the highest average reward.

Theoretical guarantees

in the form of simple expected regret bound has been provided.

The paper is in most part well-written and easy to follow.

Also the idea of using multiple instances of HOO

with different smoothness parameters seems a promising idea to deal with the

case of unknown smoothness. However I am not sure the current

version of POO really addresses the problem of unknown smoothness in online optimization. The main reason is that

the current version of proposed algorithm still needs to receive

some upper bounds of the smoothness as the input. Also regret bounds strongly depend on the value of these upper bounds

(both the rate and constant).

The paper suggests to deal with the problem of unknown \rho_\max and \nu_\max by gradually increasing the

value of smoothness parameters until they become

larger than \rho^* and \nu^*. In particular the paper suggests to increase \rho_\max and \nu_\max such that the factor D_\max \nu_\max D_\max increases with the slow (logarithmic) rate which keeps the rate of regret

intact (up to a logarithmic factor). The problem with this approach is 1- it leads to an extra constant regret term with double exponential dependency on 1/log(1/\rho).

The term 1/\log(1/\rho ) approximately equals to 1/(1-\rho) for

\rho s close to 1 which makes this constant regret term

huge and

the whole regret bound vacuous. 2- the idea of increasing \rho and \nu, combined with doubling trick, could be already applied to the

original HOO algorithm without any need for parallelization. In fact I think by using doubling trick with increasing \rho and \nu one could achieve a better regret than what we can get with POO (the constant term would be smaller and the dependency on log(n) would be quadratic instead of cubic

in the case of POO with increasing \rho_{\max}). Therefore I am not sure the new algorithm is the best way to deal with the problem of unknown smoothness since the standard methods (e.g., using doubling trick with increasing parameters) seems to provides us with a better solution.
Summary: The problem of bandit optimization of a noisy function without an explicit knowledge of smoothness is considered. However the proposed algorithm still requires some estimate on the upper bound of smoothness parameters to be efficient. Otherwise the algorithm may dramatically slow down.

Author Feedback
Author rebuttal: R2:
We agree that the current version of the algorithm requires some upper bounds on the smoothness (this point was made clear). However we believe that there is no algorithm that could optimize (achieving the optimum rate) any function in the class of functions we consider without requiring an upper bound on the smoothness defined in a way or another. This point will be made more precise in the revised version. Also, the precise dependence in the bounds \rho_max and \nu_max is clearly made explicit in Theorem 1 (and the proof for the numerical values). We disagree with the reviewer that those coefficients appear in the rate. They only appear as a multiplicative factor. The rate is (neglecting log factor in n) n^{-1/(d+2)}, where d is the smallest near-optimality dimension.

We are aware that the 1/(1-\rho) dependency makes the optimization difficult for values of \rho close to 1 (extremely difficult problems) and that is why we made this dependency explicit.

Nevertheless we disagree that a double trick applied to HOO could lead to good regret: We argue that using increasing values of \rho and \nu at each iteration of the doubling phase would reduce dramatically the performance. First, just using an upper bound of \rho^* in HOO is not enough as the performance depends on how close from \rho^* the \rho is chosen (the near optimality dimension can be dramatically impacted by the choice of a \rho too large). In other words, a singular HOO instance with a \rho > \rho^* is not enough. A second condition for \rho to guarantee low regret is that \rho is close to \rho^*, which is not achieved with the simple doubling trick solution you suggest.

Consequently, using a unique HOO instance for increasing values of \rho is not enough as the performance of each doubling phase would decrease too much and not guarantee a good regret. Instead, we believe that keeping track of several HOO instances rather than an unique with increasing \rho is important. The instance using the largest \rho can be dramatically outperformed by another one using a smaller \rho. Also, just as like a large \rho impacts the regret rate, a large \nu also impact the regret (by a constant factor) and we provide analysis for both \rho and \nu.

Choosing the number of HOO instances to launch in parallel and the grid on [0,1] to use for \rho values is not trivial (as pointed also by R3) and this is what we mainly address.

R3,R5,R8: experiments.
Our main contribution is theoretical. However, we ran experiments on difficult functions to empirically confirm that HOO with misspecified \rho performs bad and POO's performance is close to HOO's with the best \rho. We can include them in the appendix, if the reviewers find them useful.

R3: Thanks for the pointer that we will include in the revised version.

R6:
We strongly disagree in both claims.

First, we delivered results with the O() notation for clarity, but please notice that we provide all the constants in the main text. We gave finite time analysis with explicit regret. Indeed, the O() is not used to express asymptotic behavior in n but to compress expressions by hiding constant terms with respect to n while providing these constants in the main text. We completely agree with the fact that the value of these constant are important as in practice n is often low due to high cost of evaluation function. That is why our analysis does not neglect terms independent of n and provides them explicitly.

Second, while the acronyms like HOO and HCT are not spelled out, note that big part of Section 1 and the whole Section 2.2 is devoted to compare the assumptions of the prior work and relating to them. It is indeed one of our contributions in order to justify more principled near-optimality dimension (as pointed out by R8).

R9:
Our first contribution is indeed not surprising since it is based mostly on a careful check of existing proofs. Yet, we noticed a common misconception that the difficulty of the problem only lies in the metric. To our knowledge, works using both a partitioning and a metric don't use the full metric knowledge but the diameter behavior of the cells along with the partitioning. We make it clear that the metric is actually just a complex object put between the function and the partitioning, linked to the function by smoothness assumption and to the partitioning by any "well shaped" assumption. By completely removing the notion of metric we both simplified proofs and stated the result in the dimension actually characterizing the difficulty of the problem.

We disagree that our algorithm is an application of the doubling trick. First, the doubling trick implies to throw away computation whereas notice that our algorithm never throws away any samples. Instead, it makes use of all the information collected so far. Moreover, choosing the number of HOO instances to launch in parallel as well as the grid to use is not trivial and this is what we address.